# An Update on Physiopathological Roles of Akt in the ReprodAKTive Mammalian Ovary

**DOI:** 10.3390/life14060722

**Published:** 2024-06-02

**Authors:** Carlo Giaccari, Sevastiani Antonouli, George Anifandis, Sandra Cecconi, Valentina Di Nisio

**Affiliations:** 1Department of Environmental Biological and Pharmaceutical Sciences and Technologies (DiSTABiF), Università degli Studi della Campania “Luigi Vanvitelli”, 81100 Caserta, Italy; carlo.giaccari@igb.cnr.it; 2Department of Obstetrics and Gynaecology, Faculty of Medicine, School of Health Sciences, University of Thessaly, 41334 Larisa, Greece; arella_935@hotmail.com (S.A.); ganif@uth.gr (G.A.); 3Department of Life, Health, and Environmental Sciences, Università dell’Aquila, 67100 L’Aquila, Italy; 4Department of Gynecology and Reproductive Medicine, Karolinska University Hospital, SE-14186 Stockholm, Sweden; valentina.di.nisio@ki.se; 5Division of Obstetrics and Gynecology, Department of Clinical Science, Intervention and Technology, Karolinska Institutet, SE-14186 Stockholm, Sweden

**Keywords:** PI3K/Akt pathway, ovary, folliculogenesis, PCOS, POF, ovarian cancer

## Abstract

The phosphoinositide 3-kinase (PI3K)/Akt pathway is a key signaling cascade responsible for the regulation of cell survival, proliferation, and metabolism in the ovarian microenvironment. The optimal finetuning of this pathway is essential for physiological processes concerning oogenesis, folliculogenesis, oocyte maturation, and embryo development. The dysregulation of PI3K/Akt can impair molecular and structural mechanisms that will lead to follicle atresia, or the inability of embryos to reach later stages of development. Due to its pivotal role in the control of cell proliferation, apoptosis, and survival mechanisms, the dysregulation of this molecular pathway can trigger the onset of pathological conditions. Among these, we will focus on diseases that can harm female fertility, such as polycystic ovary syndrome and premature ovarian failure, or women’s general health, such as ovarian cancer. In this review, we report the functions of the PI3K/Akt pathway in both its physiological and pathological roles, and we address the existing application of inhibitors and activators for the balancing of the molecular cascade in ovarian pathological environments.

## 1. Introduction

Mammalian germ cell development is regulated by a balancing of many signaling pathways and stage-dependent epigenetic modifications. Among these, the phosphoinositide 3-kinase (PI3K)/phosphatase and tensin homolog deleted on chromosome 10 (PTEN)/Akt and tuberous sclerosis (TSC)/mammalian target of rapamycin (mTOR) pathways have been extensively studied because they are critical regulators of ovarian folliculogenesis [1] and, in a broader perspective, of other pathophysiological mechanisms [2]. In fact, many processes, such as follicle quiescence/activation and survival, somatic cell proliferation/differentiation, and even oocyte meiotic maturation, appear to be regulated by Akt and its downstream targets [1,3]. These roles have been discovered mainly by evaluating in animal models the effects on female fertility caused by the deletion of specific related genes. Additionally, the need to effectively treat severe human ovarian diseases such as cancer, premature ovarian failure (POF), and polycystic ovary syndrome (PCOS) has intensified studies aimed at uncovering the impact of Akt signaling. In this review, we reported the most recent findings on these topics.

## 2. PI3K/Akt and the Control of Follicle Activation and Growth

Primordial germ cells (PGCs) are embryonic germ cell precursors that can differentiate into gametes at the end of a complex series of processes, including migration, survival, sex differentiation, and extensive epigenetic reprogramming [4]. The population of PGCs is heterogeneous because only a portion of them will complete all these steps, while the remaining will be eliminated by mechanisms not yet well understood. PGCs undergo extensive proliferation and migration to the fetal gonads, and in both sexes form a highly conserved cellular structure, named the germline cyst [5]. In females, PGCs differentiation needs extensive epigenetic reprogramming and the activation of germ cell-specific genes (e.g., *Stella*, *Deadend-1*, *Sox2*, and *Nanog*), together with the repression of the somatic cell differentiation program [6]. The activation of the Wnt3a pathway is crucial for the specification of PGCs and mesodermal cells and for enabling epiblast responsiveness to bone morphogenetic protein 4. This factor regulates the activity of several transcription factors involved in PGCs differentiative steps [7], the absence of which dramatically affects global gene expression and epigenetic reprogramming [8].

In the developing mouse fetal ovary, PGCs differentiate into primary oocytes by entering progressively in meiosis I and, when surrounded by one layer of flattened pre-granulosa cells (preGCs), form the primordial follicles. The number of germ cells undergoes significant reduction during the transition from meiotic prophase I to primordial follicle formation, and this decrease persists until the end of gestation. By this stage, almost 85% of oocytes are eliminated through mechanisms that have been partially elucidated in recent studies [9,10]. The final size and persistence of the primordial follicle pool will determine the fertile lifespan and timing of menopause for each female [11]. This pool represents the ovarian reserve, in which follicles remain quiescent under the action of regulative factors produced by both germ and somatic cells throughout life, unless lost by atresia or recruited for maturation and ovulation [12]. Very few follicles (<1%) will complete their development undergoing ovulation, and this number can be dramatically reduced further by alteration of cellular homeostasis caused by internal and/or external factors. For example, exposure to environmental ovotoxicants, such as xenobiotics, induces aberrant primordial follicle activation in an attempt to compensate for the consequent follicle atresia [13,14,15].

The process by which quiescent follicles are recruited into the growth phase is called primordial follicle activation and is characterized by somatic cell morphological changes and continuous proliferation. This process occurs cyclically throughout a woman’s life until its physiological depletion in coincidence with menopause [16]. It is difficult to precisely quantify in vivo the pool of primordial follicles, but the level of anti-Müllerian hormone in the blood can give an approximate indication of its size [17].

In the adult ovary, primordial follicles are maintained quiescent, although mechanisms preserving their survival and oocyte quality are functionally active. The regulation of dormancy/activation requires mainly two signaling pathways: the Kit ligand/Kit receptor (KL/c-Kit) pathway in preGCs and the PI3K/Akt/forkhead box O3 (FOXO3a) pathway in oocytes (Figure 1) [18,19]. In mice, c-Kit is expressed in PGCs at embryonic day 7.5 (E7.5), while KL is expressed in somatic cells along their migration into genital ridges, with the aim of assuring their survival. If constitutively active, c-Kit causes ovarian failure due to massive primordial follicle activation, while c-Kit inactivation within oocytes determines apoptosis [20]. In mice, KL not only prevents anomalous migration in extragonadal sites but is the signal that initiates follicle activation [6,21]. KL is produced as a membrane-bound ligand able to release a soluble form after proteolytic cleavage. In the absence of the membrane-bound KL, female mice have a reduced number of PGCs and are sterile, suggesting that the membrane-bound, but not the soluble form, is essential for PGCs survival. Recently, Mi et al. [22] showed that the secretion of KL can be stimulated also by the binding of hepatocyte growth factor (HGF) to its receptor c-Met on granulosa cells (GCs). Interestingly, Yang et al. [23] demonstrated that HGF is present in the secretoma of human umbilical cord mesenchymal stromal cells, and together with miR-146a-5p and miR-21-5p foster primordial follicles activation. This finding could be of interest in the future for the treatment of patients with POF.

Although several receptor tyrosine kinases (RTKs) have been proposed to activate PI3K, c-Kit is the most widely accepted candidate in the ovary. c-Kit activates PI3K through the direct interaction with an Src Homology 2 domain on the p85 regulatory subunit of PI3K [24]. The binding of KL to c-Kit enhances oocyte PI3K activity and the phosphorylation of protein kinase B/Akt. As in other cells, Akt needs to be firstly recruited to the plasma membrane through the binding of its PH domain to phosphatidylinositol-3,4,5-trisphosphate (PIP3), then is phosphorylated at threonine 308 (Thr308), at serine 473 (Ser473) by the PI3K-dependent kinase 1 (PDK1), and by the mammalian target of rapamycin complex 2 (mTORC2), respectively. Following dissociation from the plasma membrane, Akt phosphorylates several cytoplasmic and nuclear substrates regulating cell cycle and survival, including FOXO3 and the mammalian target of rapamycin complex 1 (mTORC1). Akt covers three main roles in cellular regulation, as follows: (1) promotes cell survival by activating prosurvival proteins such as B-cell lymphoma 2 (Bcl2) and by inhibiting proapoptotic proteins such as Bcl2 Associated X, Apoptosis Regulator (BAX), Bcl2 associated agonist of cell death (BAD), and p53, caspase 7 and 9 activity; (2) stabilizes Cyclin D1 and c-Myc; and (3) promotes cell growth by phosphorylating and inhibiting the complex formed by tuberous sclerosis 1 (TSC1) and 2 (TSC2) [6,25]. Phosphorylation of TSC2 causes the activation of mTORC1 which, in turn, phosphorylates ribosomal protein S6 and promotes primordial follicle development. These effects can be counteracted by the anti-Müllerian hormone, a marker of ovarian reserve usually secreted by developing follicles [26]. While mTOR signaling has a key role in primordial follicle activation [27], a recent paper demonstrated that the inactivation of mTORC1 in oocytes via deletion of the adaptor regulatory protein Raptor dramatically affects somatic but not germ cell differentiation. Indeed, a mouse model with the depletion of Raptor shows follicular development and female fertility similar to the wild-type one. This phenotype is probably due to a compensatory mechanism involving the improvement of the PI3K pathway. Moreover, the overactivation of mTORC1 in the preGCs leads to the interruption of oocyte quiescence through the binding of c-Kit by KL and the activation of the PTEN/PI3K/Akt/FOXO3 pathway [28].

Another important target of Akt is the transcription factor FOXO3a, a protein that induces cell cycle arrest and apoptosis. When it is present in the nucleus, primordial follicle activation is inhibited and follicular quiescence maintained; by contrast, its translocation to the cytoplasm following Akt-dependent phosphorylation coincides with FOXO3a inactivation (Figure 1). FOXO3a inhibits the production of Bone morphogenetic protein 15 while enhancing p27kip1 expression, thereby maintaining a negative control on cell proliferation. Mice carrying the deletion of the FOXO3a gene showed enhanced follicular activation, quiescent pool depletion, and, finally, infertility [29]. In humans, since this nuclear-to-cytoplasmic shift occurs in postnatal but not fetal ovaries, the different localization could help identify the pool of primordial follicles determining the length of a woman’s fertile lifespan [30]. By contrast, in large mammals, no specific localization of nuclear or cytoplasmic FOXO3a has been recorded in primordial or primary oocytes [31]. Although it is still unclear if Akt is able or not to modulate TP63 (a p53 family member) functions and the response to DNA damage, it is well known that FOXO3a inactivation affects the primordial oocyte apoptosis and DNA repair processes [32,33].

Akt inactivation is usually driven by PTEN, which dephosphorylates membrane PIP3, and by two phosphatases (pT308: protein phosphatase 2A (PP2A); pSer473: PH domain leucine-rich repeat protein phosphatases (PHLPP)), respectively. In the mammalian ovary, PTEN is specifically expressed in the oocyte, and its deletion triggers excessive primordial follicle activation into the growing phase [34]. In pig and sheep fetal and neonatal ovaries, PTEN is expressed in primordial and primary follicles, confirming its participation in follicular assembly and growth. If PTEN is inactivated in GCs, Akt increases cell proliferation and decreases cell apoptosis, thus enhancing folliculogenesis and fertility in mice [35,36]. A recent paper shows that PTEN downregulation stimulates the transformation of PGCs into embryonal germ cells that, in turn, promote the development of teratoma, the most common mediastinal germ cell neoplasms [37]. In the absence of this phosphatase, primordial oocytes have the PI3K/Akt signaling constitutively active, and the global uncontrolled activation of primordial follicles results in POF. In the human ovary, recently Albamonte and collaborators [30] reported that PTEN was undetectable by immunohistochemistry in both germ and somatic cells in fetal, pubertal, and adult ovaries, a result that is in contrast with a previous report demonstrating that PTEN was necessary to regulate activation of primordial follicles [38]. The reason for this discrepancy is still unclear. In any case, it is evident from all these results that Akt activity is finely controlled by a balance between PTEN and PI3K, and only proper regulation assures the maintenance of fertility [1,6].

Another crucial protein is PDK1, which phosphorylates and activates Akt and ribosomal protein S6 kinase (S6K), a substrate and major effector of mTORC1. Its deletion leads to POF [39].

Finally, it is of interest to mention the hypothesis of Dai et al. [18], who proposed the existence of distinct age-dependent mechanisms regulating follicle activation in mouse ovaries. According to this hypothesis, the first wave of follicle activation recorded in the postnatal ovary is exclusively triggered by oocyte-dependent signaling based on precocious activation of PI3K signaling and meiosis initiation and occurs in the absence of preGCs-dependent KL secretion. By contrast, in the adult mouse ovary, follicular waves are dependent on KL and on Follicle Stimulating Hormone (FSH), which are both able to trigger follicular activation and growth via stimulation of PI3K/Akt and downstream FOXOs. If confirmed in humans, this finding reinforces the idea that the first wave of primordial follicle activation might be involved in determining the onset of puberty.

During folliculogenesis, gonadotropic hormones—FSH and Luteinizing Hormone (LH)—together with many paracrine factors determine differentiation and atresia of GCs [40]. FSH stimulates GCs proliferation also by activating PI3K/Akt and downstream targets Cyclin D2 and mTOR1 [41], as well as mitogen-activated protein kinase (MAPK)3/1. This last finding supports the use of specific kinase inhibitors to safeguard ovarian reserve [42]. In contrast, oxidative stress inhibits the PI3K/Akt pathway, and the consequent increase in GCs apoptosis can determine precocious ovarian aging and female subfertility [43,44]. Recently, Wei et al. [45] demonstrated that the Src homology 2 domain-containing protein tyrosine phosphatase 2 (SHP2) has a role in the balancing of GCs proliferation and apoptosis because it impairs FSH-dependent PI3K activation since its deletion promotes follicular growth. As in other organs, in the ovary SHP2 activation is induced by oxidative stress, which suppresses PI3K/Akt activity by promoting the binding between SHP2 and the regulatory subunit p85 of PI3K.

## 3. PI3K/Akt Role during Oocyte Maturation and Fertilization

Complementary to its pivotal role during folliculogenesis, the PI3K/Akt pathway is also a key regulator of mammalian oocyte maturation and developmental competence [3,6]. Indeed, in female germ cells, Akt-dependent mechanisms have been proven to drive oocyte maturation at the nuclear and cytoplasmic levels through the inhibition of this pathway with different classes of inhibitors, although in some cases this is species-specific [46]. This large body of literature shows how the interference with the PI3K/Akt/PTEN pathway can impair germinal vesicle breakdown (GVBD) in murine and porcine immature oocytes [47,48,49,50,51,52,53] and the progression up to metaphase II (MII) stage in porcine, bovine, and murine oocytes [48,49,54,55,56,57].

During the process of meiotic resumption, the immature GV oocytes undergo a complex interplay of activatory/inhibitory molecular mechanisms that are mainly focused on the decrease in cyclic AMP (cAMP) ooplasmic concentration. These mechanisms are strictly dependent on the redundant signaling promoted by the LH surge at ovulation, culminating in the production of EGF-like factors and downregulation of the meiotic inhibitory network. This process, mediated by C-type natriuretic peptide (CNP) and cyclic guanosine monophosphate (cGMP), results in the final activation of the CDK1/Cyclin B complex, i.e., Maturation Promoting Factor (MPF). Akt exerts its function through the inhibitory phosphorylation of the kinases Wee1 and Myt1 [58], and the activation of the phosphatase CDC25 [59] that removes the two inhibitory phosphate residues (Thr14 and Tyr15) from CDK1. The Akt-dependent downstream signaling is therefore crucial for the initiation of the GVBD process in mice, bovine, starfish, and zebrafish, but not essential in porcine oocytes [50,60,61,62,63,64,65]. Interestingly, a recent study elegantly demonstrated the effect of SUMOylation on the expression and localization of phosphorylated-Akt (pAkt)-Ser473 and its role in meiotic resumption in vitro. pAkt-Ser473 was detected in the ooplasm of GV-stage oocytes, being relocated on the nuclear membrane during GVBD and subsequently scattered in the cytoplasm. This process was affected when oocytes were in vitro matured in the presence of gingkolic acid, a potent inhibitor of SUMOylation processes [66]. Additionally, the inhibition of SUMOylation affected the pAkt-Ser473-dependent process of Cyclin D1 redistribution in the nucleus [66].

The role of Akt during oocyte maturation is not only restricted to GVBD but is also fundamental to the molecular and structural regulation of the spindle assembly and cytoplasmic cytoskeletal dynamics. In fact, the Akt-dependent activation of the phosphodiesterase 3A (PDE3A) not only results in cAMP hydrolysis but also in the stimulation of the plectin-1 binding on the intracellular cytoskeleton network [60,67]. Even more important is the role of Thr308- and Ser473 pAkt for the successful formation and stabilization of meiotic spindles at the MII stage [50,68,69]. The different localization of these two phosphorylated forms on the spindle has been linked to the distinct role of Akt as an mTORC1/2 upstream regulator. In fact, while pAkt-Thr308 is detected mainly at the spindle poles, corresponding to the so-called microtubule-organizing centers (MTOCs), pAkt-Ser473 localizes along the spindle microtubules [50,68,69]. This determines a differential localization of the two mTORC isoforms: if mTORC2 colocalizes with pAkt-Thr308 in the MTOCs, mTORC1 is structurally and functionally located at spindle microtubules together with pAkt-Ser473, the ribosomal protein S6, and the eukaryotic translation initiation factor 4E-binding protein 1. Thus, Akt regulates mTORCs and downstream factors activities [3,70]. Additionally, the constitutive activation of Akt in *Pten* knock-out mice is related to the mRNA translation of factors (as *Tpx2*, *Dazl*, or *Il7*) involved in spindle assembly, chromosome segregation, and, most importantly, the acquisition of developmental competence [71,72].

The PI3K/Akt pathway is also a key player during and soon after fertilization. In fact, Akt activity rises during meiotic maturation, reaches a plateau during MI/MII transition, and remains elevated during MII arrest, fertilization, and early embryonic development [3,46,73]. The high Akt activity is connected to CC expansion because, after the LH surge, CCs start producing hyaluronic acid (HA) in a PI3K/Akt-dependent manner [74]. In vitro inhibition of the PI3K/Akt pathway in porcine cumulus-oocyte complexes led to a dramatic reduction in HA synthase 2, therefore impairing CC expansion [75,76].

Also, the high activity of Akt during the first stages of embryo development, by upregulating anti-apoptotic signals in fertilized murine oocytes [77] and by nuclear translocation of its phosphorylated forms, is necessary to maintain proliferation stimuli through mTOR signaling-related mechanisms [78,79]. Particularly, the function of pAkt-Ser473 has been identified to be involved in zygote genome activation processes and blastomeres proliferation in several mammalian species [80,81,82].

## 4. PI3K/Akt Pathway in Reproductive Pathologies

Due to the numerous relevant roles of the PI3K/Akt pathway in ovarian physiology, any alteration in the finetuning of this central molecular signaling can lead to reproductive pathologies, such as Polycystic Ovary Syndrome (PCOS) and POF. In this section, we will walk through the in situ and endocrinological effects of PI3K/Akt pathological dysregulation, the main modulators of this pathway (summarized in Table 1), and the possible treatments against it (reported in Table 2).

### 4.1. Polycystic Ovary Syndrome

PCOS represents a complex endocrine disorder affecting women of reproductive age. Its complexity is manifested through a combination of reproductive and metabolic features, as its key clinical characteristics are hyperandrogenism, anovulation, polycystic ovaries, insulin resistance (IR), and type 2 diabetes mellitus [83,84]. GCs have a crucial role in folliculogenesis, and their alterations generate reproductive disorders. As previously reported, during folliculogenesis, the PI3K/Akt pathway is crucial in maintaining the balance between GCs growth and apoptosis through modulation of pro-apoptotic genes such as *BAX*, *CASP9* (caspase 9), *CASP3* (caspase 3), and *FOXO1*. In GCs from PCOS patients, the expression of these genes is stimulated, while the expression of pPI3K, pAkt, and *BCL2* is diminished. This anomalous apoptotic stimulus may be attributed to excessive reactive oxygen species (ROS) production and mitochondrial dysfunctions [85].

IR and hyperinsulinemia affect up to 70% of women with PCOS [86]. Under physiological conditions, insulin activates the PI3K/Akt pathway in GCs and stimulates glucose metabolism. Although insulin’s action on mitogenesis/steroidogenesis is preserved in the GCs of PCOS, the metabolic effects of this hormone on glucose uptake, glycogen synthesis, and lactate production are severely compromised [87]. The High mobility group box 1 protein (HMGB1) emerges as a potential contributor to IR development in PCOS patients. The abundance of HMGB1 is notably elevated in the follicular fluid of PCOS patients with IR, potentially resulting from uncontrolled autophagy. Furthermore, treatment with HMGB1 reduces insulin-induced Akt phosphorylation, hampers GLUT4 translocation from the cytoplasm to the cell membrane in response to insulin, and slows down glucose uptake in GCs from non-PCOS women [88]. PTEN, which negatively regulates the PI3K/Akt pathway, is highly expressed in the GCs of PCOS patients and is associated with elevated insulin levels in follicular fluid [89]. Ubiquitin carboxyl-terminal hydrolase 25 (USP25) is capable of deubiquitinating PTEN in GCs. This enzyme is highly expressed in both PCOS patients and mouse models: its depletion can ameliorate IR symptoms, while its absence in GCs enhances the activity of the PI3K/Akt signaling pathway. Additionally, the expression of GLUT4 and IRS-1 significantly increases [90]. Another potential contributor to IR is the microRNA miR-133a-3p, which is overexpressed in the GCs of PCOS patients. The majority of miR-133a-3p targets are linked to the PI3K/Akt pathway and play crucial roles in processes such as glucose and lipid metabolism. Notably, the elevated expression of this microRNA negatively modulates the activity of the PI3K/Akt signaling. Specifically, miR-133a-3p interferes with glucose transporter type 4 (GLUT4) activity and activates proteins such as p-glycogen synthase kinase 3 (GSK3)-β and pFOXO1, thereby contributing to the regulation of IR in the ovaries [91]. Moreover, it is well-established that miRNAs contained in follicular fluid exosomes play a crucial role in fertilization mechanisms and impact the proliferation of GCs [92]. From a recent study, it appears that PTEN may be negatively regulated by the microRNA miR-18b-5p, which is downregulated in the GCs and follicular fluid of patients with PCOS. Increasing miR-18b-5p in exosomes of follicular fluid resulted in GC proliferation and decreased apoptosis. Thus, the overexpression of miR-18b-5p mitigated sex hormone imbalances and reduced pathological damage in the ovaries of PCOS rats, indicating that the miR-18b-5p/PTEN axis may regulate the PI3K/Akt/mTOR pathway in the progression of PCOS [93].

Despite the limitations of the in vitro approach on human cells or in vivo with animal models, these methods remain crucial for uncovering new mechanisms that cannot be directly assessed in humans, whose studies are based mainly on endocrinological and general approaches that exclude the analysis of invasive ovarian biopsies. Rats with PCOS treated with a high-fat diet tend to exhibit a more severe phenotype marked by a significantly reduced kisspeptin protein content in GCs. Indeed, kisspeptin reduces OS and apoptosis by activating PI3K/Akt and ERK1/2 pathways. In KGN cells, a reduction in pro-apoptotic factors, such as BAX and caspase-3, and an increase in anti-apoptotic factors, like Bcl2, are associated with high levels of kisspeptin. Conversely, the reduction in kisspeptin increases the expression of cleaved caspase-3 and BAX while decreasing the expression of Bcl2 [94]. Studies using a rat model of PCOS have revealed that inhibiting the expression of the long non-coding RNA HOTAIR, which is overexpressed in PCOS, can reverse these hormonal imbalances. For example, PCOS-afflicted rats often manifest irregular estrus cycles, but the inhibition of HOTAIR helps to regulate them. This syndrome can lead also to structural abnormalities in rat ovaries characterized by ovarian fibrosis and a reduction in the number of antral and preantral follicles. The inhibition of HOTAIR expression partially restores normal ovarian morphology, including an increase in the number of GCs that occurs through the reversal of the apoptotic genes’ expression. The authors reported that the in vivo silencing of HOTAIR in PCOS rats restored the physiological hormonal balance—by decreasing serum levels of testosterone, estradiol, and LH, and increasing FSH. Moreover, both in vivo and in vitro the blockade of HOTAIR functions resulted in a reduction in insulin-like growth factor 1 (*IGF1*) gene expression and Akt phosphorylation level [95].

### 4.2. Premature Ovarian Failure

POF, also known as primary ovarian insufficiency or early menopause, is a medical condition that affects the reproductive health of women. It is characterized by the loss of normal ovarian function before the age of 40, leading to a decline in fertility and the onset of symptoms associated with hormonal imbalances. Being the PI3K/Akt pathway fundamental in folliculogenesis, its alteration could lead to ovarian follicle dysfunction or the reduction in primordial follicles, typical of POF [96].

POF is characterized by an increase in pro-inflammatory interleukins such as IL-1β, IL-4, and IL-6 in GCs. Specifically, IL-4 is capable of altering the PI3K/Akt pathway, stimulating the expression of pro-apoptotic proteins in GCs [97]. The typical symptoms of POF can also be induced by substances such as Cyclophosphamide (CTX), used in therapies against cancer or autoimmune diseases. The application of this substance in mouse models leads to the alteration of the Rictor/mTORC2/Akt/FOXO3a signaling axis through the downregulation of pAkt and pFOXO3a, confirming the critical role of Akt in the progression of this pathology [98]. Treatment with the chemotherapeutic agent Cisplatin also induces an accumulation of ROS in GCs and leads to the typical symptoms of POF. This may be correlated with the reduction in Zinc (Zn) observed after Cisplatin treatment, which acts as an antioxidant by promoting the activation of Akt. The activated Akt is capable of entering mitochondria and phosphorylate GSK3β, whose inhibition restricts ROS levels [99].

In the pathological mechanisms of POF, miRNAs are also implicated. The transmembrane protein Klotho is a regulator of the PI3K/Akt pathway, and its expression is reduced in the GCs of POF. This protein is regulated by various miRNAs, such as miR-497-3p. In a study involving KGN cells treated with CTX, it was observed that this miRNA is overexpressed and, through the downregulation of Krüppel-like factor 4 (KLF4), it reduces the expression of Klotho [100]. In a study with a rat model of POF, obtained through treatment with 4-vinylcyclohexene diepoxide, an up-regulation of miR-190a-5p was identified. Using bioinformatic tools, the PH domain and leucine-rich repeat protein phosphatase 1 (PHLPP1) have been identified as a target of this miRNA, and its expression is reduced in POF rats. PHLPP1 is already known as a phosphatase of Akt, and its reduced expression is consistent with the increase in phosphorylated Akt and FOXO3a, which leads to the hyperactivation of primordial follicles in POF [101,102].

**Table 1 life-14-00722-t001:** Modulators of Akt pathway in reproductive pathologies.

Modulator	Level	Effect on PI3K/Akt Related Proteins	Reproductive Disease	Research Subjects	References
HMGB1	Increased	↓ GLUT4 ↓ pAkt	PCOS	Human	[89]
HOTAIR	Increased	↓ Bcl2 ↑ BAX ↑ IGF1 ↑ pAkt	PCOS	Rat model	[84]
IL-4	Increased	↓ Bcl2 ↑ Caspase-3 ↑ Caspase-9 ↑ BAX ↑ pAkt	POF	Human	[98]
kisspeptin	Reduced	↓ Bcl2 ↑ Caspase-3 ↑ BAX ↓ pAkt	PCOS	Human Rat model	[85]
miR-18b-5p	Reduced	↑PTEN	PCOS	Human Rat model	[93]
miR-133a-3p	Increased	↓ GLUT4 ↑ pGSK3β ↑ pFOXO1 ↓ pAkt	PCOS	Human	[88]
miR-190a-5p	Increased	↓ PHLPP1 ↑ pFOXO3a ↑ pAkt	POF	Rat model	[99,100]
miR-497-3p	Increased	↓ KLF4 ↓ Klotho ↓ pAkt	POF	Human	[101]
USP25	Increased	↓ GLUT4 ↓ IRS-1	PCOS	Human Mouse model	[91]
Zinc	Reduced	↓ pAkt ↓ pGSK3β	POF	Rat model	[97]

Abbreviations: BAX, Bcl-2 associated X protein; Bcl2, B-cell lymphoma 2; GLUT4, glucose transporter type 4; HMGB1, high mobility group box 1; HOTAIR, HOX transcript antisense intergenic RNA; IGF1, insulin-like growth factor 1; IL-4, interleukin 4; IRS-1, insulin receptor substrate 1; KLF4, krüp-pel-like factor 4; pAkt, phosphorylated protein kinase B; PCOS, polycystic ovary syndrome; pFOXO1, phosphorylated forkhead box O1; pFOXO3a, phosphorylated forkhead box O3a; pGSK3β, phosphorylated glycogen synthase kinase 3 β; PHLPP1, PH domain and leucine-rich repeat protein phosphatase 1; POF, premature ovarian failure; PTEN, phosphatase and tensin homolog deleted on chromosome 10; USP25, ubiquitin-specific protease 25; ↑ increased expression; ↓ decreased expression.

### 4.3. Potential Treatments for Reproductive Pathologies Involving the PI3K/Akt Pathway

Recent studies are focused on the development of new therapies to improve the conditions of PCOS and POF patients. For example, the use of berberine, an isoquinoline compound derived from plants, in rat models of PCOS improves ovarian morphology by increasing the number of GCs and the presence of oocytes in follicles. In fact, berberine positively influences the PI3K/Akt signaling pathway, leading to a significant increase in pAkt levels and GLUT4 expression in PCOS rats. This suggests that berberine may protect against PCOS-associated IR by restoring the activation of the PI3K/Akt pathway [103]. The use of selenium nanoparticles (SeNPs) and metformin (MET), either individually or in combination, could be a potential treatment for PCOS-related IR. In a study conducted on a rat model, treatment with SeNPs or MET increases the proliferation of GCs and estradiol production, together with enhancement of mitochondrial activity, likely due to the antioxidant action of both SeNPs and MET. The effect of these substances is probably attributable to the activation of the PI3K/Akt pathway, as the treatment increases the expression of PI3K and Akt [104]. Also, Melatonin, an antioxidant produced physiologically, has been observed to have therapeutic effects on folliculogenesis and oocyte maturation in PCOS. This substance is capable of maintaining mitochondrial membrane potential by directly acting in the removal of ROS. The effect of Melatonin is attributed to the overexpression of sirtuin 1 (SIRT1), which can increase levels of pAkt. As evidence of the direct interaction between SIRT1 and the Akt pathway, in SIRT1 knockdown cells levels of pAkt and membrane potential are compromised despite Melatonin treatment [105]. The growth hormone (GH) reduced ROS production and the apoptosis rate in GCs. GH indirectly influences the production of IGF-1 by activating the IRS receptor and the Akt pathway. In fact, in vivo administration of GH partially reverses the effects of ROS by increasing the expression of PI3K, Akt, and Bcl2, while decreasing the expression of FOXO1, BAX, caspase-9, and caspase-3 [85].

The use of small extracellular vesicles derived from embryonic stem cells (ESC-sEVs) appears to have positive effects on ovarian function. In a murine model of POF, the ESC-sEVs can enhance hormonal serum levels, reduce apoptosis, and increase GCs proliferation by suppressing the expression of pro-apoptotic proteins and increasing anti-apoptotic ones. The activated regulatory mechanism may involve the PI3K/Akt pathway, as GCs treated with ESC-sEVs demonstrate an increase in Akt phosphorylation [106]. Similar results have been found in a study on KGN cells, identifying the treatment with human umbilical cord mesenchymal stem cell-derived extracellular vesicles (hucMSCEVs) as a potential therapy to improve POF conditions through the regulation of the PI3K/Akt pathway. A miRNA may play a pivotal role in this mechanism. It has been observed that miR-126-3p, present in hucMSCEVs, can transfer into Primary Rat Ovarian GCs, suppressing apoptosis by downregulating Phosphoinositide-3-Kinase Regulatory Subunit 2 (PIK3R2) and activating the PI3K/Akt pathway [107,108]. Resveratrol also appears to improve the conditions of POF. Following treatment with this molecule, GCs exhibit activation of the PI3K/Akt pathway and a reduction in oxidative stress. Moreover, through the regulation of pro- and anti-apoptotic proteins, resveratrol leads to a decrease in apoptotic events [109]. 

Altogether, these results confirm that the alteration of the PI3K/Akt pathway emerges as a crucial node in complex reproductive syndromes such as PCOS and POF, influencing gene regulation, insulin resistance, and metabolic homeostasis (Table 1 and Table 2). Research into therapeutic solutions, through the identification of regulatory microRNAs and the exploration of innovative treatments, provides new directions in clinical management, indicating potential approaches to enhancing the quality of life for women affected by PCOS and POF. 

**Table 2 life-14-00722-t002:** Treatment of reproductive pathologies involving the PI3K/Akt pathway.

Treatment	Effect on PI3K/Akt Related Proteins	Reproductive Disease	Research Subjects	References
Berberine	↑ pAkt ↑ GLUT4	PCOS	Rat model	[103]
ESC-sEVs	↑ pAkt	POF	Mouse model	[106]
GH	↑ IGF1 ↑ PI3K ↑ Akt ↑ Bcl-2 ↓ Caspase-3 ↓ Caspase-9 ↓ BAX	PCOS	Human	[85]
hucMSCEVs	↓ PIK3R2 ↑ pAkt	POF	Human	[107,108]
Melatonin	↑ SIRT1 ↑ pAkt	PCOS	Human	[105]
Resveratrol	↑ pPI3K ↑ pAkt ↑ Bcl-2 ↓ Caspase-3 ↓ BAX	POF	Rat model	[109]
SeNPs and MET	↑ PI3K ↑ Akt	PCOS	Rat model	[104]

Abbreviations: BAX, Bcl-2 associated X protein; Bcl-2, B-cell lymphoma 2; ESC-sEVs, small extracellular vesicles de-rived from embryonic stem cells; GH, growth hormone; GLUT4, glucose transporter type 4; hucMSCEVs, human umbilical cord mesenchymal stem cell-derived extracellular vesicles; IGF1, insulin-like growth factor 1; MET, metformin; pAkt, phosphorylated protein kinase B; PCOS, polycystic ovary syndrome; PI3K, phosphoinositide 3-kinase; PIK3R2, phosphatidylinositol-3-kinase regulatory subunit 2; POF, premature ovarian failure; pPI3K, phosphorylated phosphoinositide 3-kinase; SeNPs, selenium nanoparticles; SIRT1, silent information regulator sirtuin 1. ↑ increased expression; ↓ decreased expression.

## 5. PI3K/Akt Pathway in Ovarian Cancer

The onset of ovarian cancer (OC) is molecularly connected to mutagenesis of specific genes that can either regulate the expression of cell cycle/division mechanisms—generally known as proto-oncogenes, e.g., *K-ras*, *c-myc*, and *c-erbB-2*—as well as of genes that are involved in the cell survival pathways—downregulation of tumor suppressors, e.g., *TP53*, *BRCA1*, *BRCA2*, and *PIK3CA* [110]. *PIK3CA* is a key modulator of the PI3K/Akt pathway for the promotion of cell survival, proliferation, and glucose metabolism [111]. The close relation of the PI3K/Akt/mTOR pathway to the MAPK-dependent mechanisms increases even more the importance of this molecular balance and their targeting in patients diagnosed with OC [112]. In the following section, we will unravel the main roles of Akt regulation in OC pathogenesis and development (summarized in Figure 2), together with the up-to-date cancer treatment strategies currently in clinical use or trial to counteract Akt’s role in OC physiopathology.

### 5.1. PI3K/Akt Role in OC Pathogenesis

As previously discussed, Akt is a main actor in the PI3K/Akt/mTOR signaling pathway that plays a key role in the regulation of many mechanisms involved in OC pathogenesis, e.g., cell survival, growth and proliferation, angiogenesis, and metabolism. In this context, Akt has been found to act as both an oncogene and an oncoprotein in OC [113]. The role of *AKT* as an oncogene has been extensively described in ovarian tumors. Particularly, the oncogenic activity of *AKT* has been recognized when a point mutation in the N-terminal plekstrin homology domain is detected [114]. This mutation can be thereafter activated despite the upstream PI3K signaling normal regulation. Mutations in *AKT1*, together with amplification and copy number gain in *AKT2*, have been described in GCs tumors [115], ovarian serous neoplasm [116], and primary OCs [117]. Specifically, the *AKT1* mostly mutated isoform is found in the E17K and Q79K sites [118], while almost 40% of OC cases report an amplification of *AKT2* [119]. Interestingly, 35% of high-grade serous OC present a constitutive upregulation of the PI3K/Akt/mTOR pathway [120], and an additional 10–40% link its hyperactivation due to the deregulation of the upstream Rat sarcoma (RAS) signaling [121]. In addition, the dysregulation of both *PTEN* gene expression—through both gene copy number variation and miRNAs-dependent gene expression regulation—and pathogenic variants have been associated with OC increased risk [122], mainly driven by the deregulation of the Akt activity and expression [123].

The main reports are predominant in the role of Akt as an oncoprotein among OC cell lines, animal models, and ovarian tumors. In fact, the dysregulation of Akt signaling through the phosphorylation of one of these kinase isoforms in over 500 OCs has been associated with decreased survival and identified as a negative prognostic marker [124,125]. Screening ovarian cancer cell lines commercially available—e.g., OVCA429, OVCA432, OVCAR3, DOV13, and SKOV3—shows the presence of one or multiple isoforms of Akt (Akt1, 2, and 3) [113]. Interestingly, the study from Cristiano and collaborators described the predominant role of Akt3 in the pathogenesis of OC, mainly due to its modulation of G2-M phase transition, and consequent cell proliferation. Through the measurements of Akt activity in vitro in the presence or absence of short hairpin RNA for Akt3 silencing, the authors were able to demonstrate that the Akt3 isoform was responsible for regulating the G2-M phase transition via the downstream Wee1 kinase pathway [113]. 

The direct and indirect inhibition of apoptosis through Akt and further anti-apoptotic factors Bcl-2 and X-linked inhibition of apoptosis protein (XIAP) activation can also play a key role in both OC onset and multidrug resistance [126]. A study on epithelial OC-derived cells and cancer cell lines—namely CaOV3 and OVCAR3—demonstrated the pivotal role of Akt1 in the Tumor necrosis factor-related apoptosis-inducing ligand (TRAIL)-induced apoptosis, through the extrinsic apoptosis-inhibition of the initiator caspase 8 and the intrinsic apoptosis-interference with Bid cleavage (truncated Bid, tBid). Therefore, the lower threshold of tBid leads to insufficient activation of BAX and Bcl-2 homologous antagonist/killer (Bak) pro-apoptotic proteins, accompanied by an inefficient mitochondrial release of cytochrome c and subsequent apoptosis activation [127]. On the other hand, the TRAIL-resistant SKOV3ip1 cell line showed full sensitivity to the apoptotic process when treated with Akt inhibitors, through the truncation of tBid and the downstream apoptotic process activation [127]. Additionally, the intrinsic apoptosis involvement in Akt-dependent OC pathogenesis has been demonstrated through the supporting role of cleaved caspase 9 and the inhibition of XIAP-dependent downregulation of this mitochondrial-related pathway in the *BRCA2* mutated PEOC1 cancer cell line [128]. In fact, cells treated in vitro with cisplatin showed downregulation of both intrinsic and extrinsic apoptotic pathways through the double inhibition of Akt and of the DNA-dependent protein kinase [128].

Overall, the reported studies indicate Akt and the close upstream or downstream regulators (PI3K and mTOR, respectively) as some of the main actors in the cell cycle promotion and apoptosis resistance of OC cells.

### 5.2. PI3K/Akt Role in OC Development

Once the onset of OC starts, the malignant cells require a high level of energy for fast cell division and survival, and as the pathology progresses, metastasis also occurs. The PI3K/Akt/mTOR dysregulation has been identified as a main pathway activated throughout these processes, mainly due to its involvement in the energy metabolism through the Akt2-mediated cross-talk with GSK3, and the PI3K/Akt/mTOR susceptibility to the AMP-activated protein kinase (AMPK). GSK3 dysregulation has been extensively described in cancer onset and development and is depicted as a key molecule among PI3K/Akt/mTOR, AMPK, and mTORC signaling [129]. Moreover, the upregulation of GSK3β in OC has also been connected to its main role in the activation of glycolysis, the main metabolic pathway supporting energy sustainability in OC [130,131]. In addition, the activation of AMPK during metabolic stress monitors the AMP/ATP ratio, is responsible for the downstream regulation of ROS-dependent fatty acid oxidation metabolism, and can suppress the mTOR pathway through the inhibition of mTORC1 [132]. The dysregulation of AMPK in OC is therefore crucial for the malignant cells, allowing their escape from the classical proliferation controls during the cell cycle [133]. In fact, treatments of OC cell lines in vitro with the hypoglycemic drug MET have been proven to reduce ovarian cancer cell growth through AMPK-Akt-dependent mechanisms [134]. The mode of action of this medical treatment consists of the activation of AMPK, which in turn inhibits Akt, thus preventing the Akt-mediated GSK3 inhibitory phosphorylation. In addition, the active GSK3 stimulates the ubiquitination of cyclin D that undergoes proteolysis and therefore causes the blockade of the cell cycle in OC cells [135]. Lastly, the upstream regulation of the PI3K/Akt pathways of the steroid receptor coactivator 3 (SRC-3) and tumor necrosis factor receptor-associated factor 4 (TRAF4) has recently been associated with OC development [136]. In vitro studies on SKOV3 and CaOV3 OC cell lines demonstrated not only the close relation between these two markers, but also the direct effect of their upregulation on cell growth, migration, and invasion through Akt-dependent mechanisms [136].

Interestingly, C57Bl6 in vivo studies on different Akt isoforms inhibition underlined the several different modes of action of Akt1, 2, and 3 on OC progression. Indeed, while the inhibition of Akt1 was significantly linked to the reduced proliferation and tumor progression of in vivo OC cells, the silencing of Akt2 increased the tumor growth, and *Akt3*-KO mice displayed an intermediate phenotype, leaning towards the stimulation of OC cell growth [137]. In addition, OC cell growth and metastasis mechanisms were reduced in vivo in the presence of cinnamaldehyde, a bioactive natural substance isolated from cinnamon bark. Its mode of action involves suppressing the epithelial-mesenchymal transition (EMT) via Akt-mediated mechanisms [138]. Similarly, in vivo treatment with Danshen-extracted Tanshinone IIA has been proven to reduce PI3K and Akt phosphorylated forms, therefore inhibiting OC cell growth and enhancing both intrinsic and extrinsic apoptosis through the caspase 3- and 8-related mechanisms [139]. The inhibition of PI3K/Akt/mTOR pathway in vitro showed the effect on decreased chemoresistance of epithelial OC cells by the use of dactolisib in combination with cisplatin, increasing both ROS production and apoptosis mechanisms, and thus decreasing rates and outcomes of EMT [140]. Likewise, patient-derived OC spheroids cultured in vitro with LY-294002—a PI3K-Akt dual kinase inhibitor—showed a significant decrease in cancer stemness and chemoresistance to cisplatin [141], and resveratrol-treated OC cells SKOV-3 displayed increased apoptosis and induced cell cycle arrest through Akt suppression in vitro [142].

In summary, the inhibition of Akt and its closely related pathway can have great effects on OC cells, both in vitro and in vivo, by targeting crucial mechanisms such as cancer cell growth, metabolism, metastasis, and apoptosis. These results depict Akt and its up- and downstream regulators as key players in the development of OC, and thus worthy of further investigation in therapeutic clinical anti-OC treatments and perspectives.

### 5.3. Possible Strategies to Counteract PI3K/Akt Role in OC

As already introduced in the previous section, targeting the PI3K/Akt/mTOR pathway is a hot topic in the development of new OC therapies, as extensively reviewed elsewhere [110,111,143,144,145,146]. In this context, several strategies have been tested to prevent OC development through pan-PI3K inhibitors, isoform-specific inhibitors for PI3K, Akt, and mTORC1 and/or mTORC2, together with the dual pan-PI3K/mTORC1/mTORC2 inhibitors that inhibit at the same time the four main PI3K isoforms, mTORC1 and mTORC2 [146]. Focusing on the Akt inhibition, thereby leading to the silencing of the downstream mTOR and cell cycle-related mechanisms activation, MK-2206, TAS-117, and perifosine (alone or in combination with paclitaxel) treatments demonstrated significant effects in vitro and in vivo on the improvement of cytotoxicity, contemporarily increasing the chemotherapeutic sensitivity to cisplatin-, irinotecan-, and carboplatin-dependent apoptosis on OC cell lines [147,148,149,150,151].

On this basis, clinical trials are currently ongoing to test miransertib, ipatasertib, uprosertib, BAY1125976, AZD5363, ARQ 751, and MSC2363318A as Akt inhibitors in several types of cancers, including OC [152,153,154]. In particular, MK-2206 has been recently investigated in OC clinical trials, as a monotherapy or in combination with chemotherapy or targeted cancer therapies. However, it was found to be clinically ineffective, partly due to the poor accrual of patients to the trial itself [155]. On the other hand, the perifosine adjuvant treatment in platinum-resistant OC cases led to 1.9 months of progression-free survival and 4.5 months of overall survival without dose limitations due to toxicity [156]. In platinum-resistant OCs, a phase Ib/II clinical study investigated the effect of afurosertib, concluding in positive results with >7 months of progression-free survival and objective response rates of 32% (RECIST criteria) and 52% (GCIC CA125 criteria) [157]. Finally, new clinical trials are now starting to recruit more platinum-resistant OC patients for the interrogation of the effectiveness of the combination treatment afuresertib-paclitaxel (PROFECTA-II; NCT04374630).

## 6. Conclusions

The PI3K/Akt pathway plays an important role in the physiopathology of the ovary, as extensively reported in the present review. In fact, the finetuning of this complex molecular signaling is necessary throughout follicle establishment and activation, and up to ovulation. By regulating pivotal processes such as cell cycle, cell proliferation, and survival, the PI3K/Akt pathway is also essential during oocyte meiotic maturation, the acquisition of developmental competence, and the first stages of embryo development. The dysregulation of PI3K/Akt-dependent signaling can lead to the initiation and development of ovarian reproductive pathologies, such as PCOS and POF, leading to female subfertility. In addition, the mutation of peculiar actors in this pathway—such as *PIK3CA*, *AKT*, and *PTEN*—are known biomarkers for the etiology and development of OC, highlighting even more the fundamental role of these genes and proteins in the molecular balance necessary for ovarian health. On this basis, the development of novel drugs targeting different nodes of this pathway—e.g., eukaryotic translation initiation factor 4E (eIF4E), p70S6 kinase (p70-S6K), and proto-oncogene serine/threonine-protein kinase (PIM)—could help the progress of anti-OC therapies. Further studies on the crosstalk of the PI3K/Akt pathway with other crucial cellular mechanisms are imperative to shed light on the possible application aimed at counteracting its pathological role.

## Figures and Tables

**Figure 1 life-14-00722-f001:**
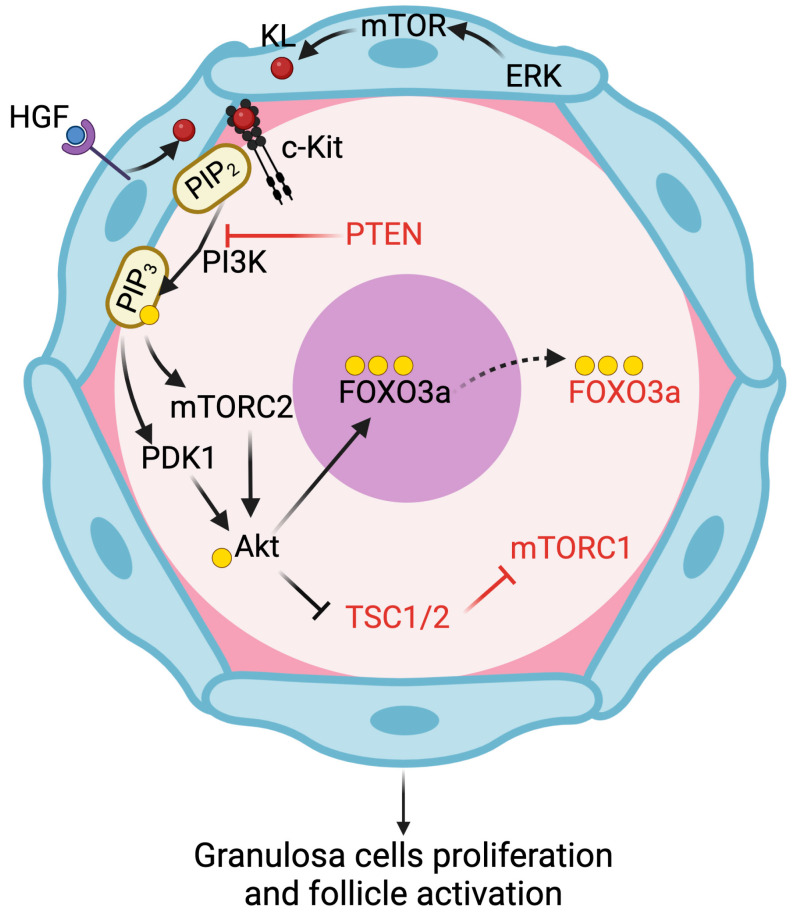
**PI3K/Akt and KL role in primordial follicle activation**. KL exploits its role mainly in granulosa cells to promote their proliferation via mTOR and binds the oocyte c-Kit membrane receptor that initiates the PI3K/Akt signaling cascade. The main active signals are reported in black, while the inhibited/inactive markers are evidenced in red. The yellow circle indicates the presence of a phosphate residue. ERK, Extracellular signal-regulated kinase; FOXO3a, forkhead box O3; HGF, hepatocyte growth factor; KL, Kit ligand; mTOR, mammalian target of rapamycin; mTORC1, mammalian target of rapamycin complex 1; mTORC2, mammalian target of rapamycin complex 2; PI3K, phosphoinositide 3-kinase; PDK1, phosphoinositide 3-kinase-dependent kinase 1; PIP_2_, phosphatidylinositol-4,5-bisphosphate; PIP_3_, phosphatidylinositol-3,4,5-trisphosphate; PTEN, phosphatase and tensin homolog deleted on chromosome 10; TSC1/2, tuberous sclerosis 1/2. Created with BioRender.com.

**Figure 2 life-14-00722-f002:**
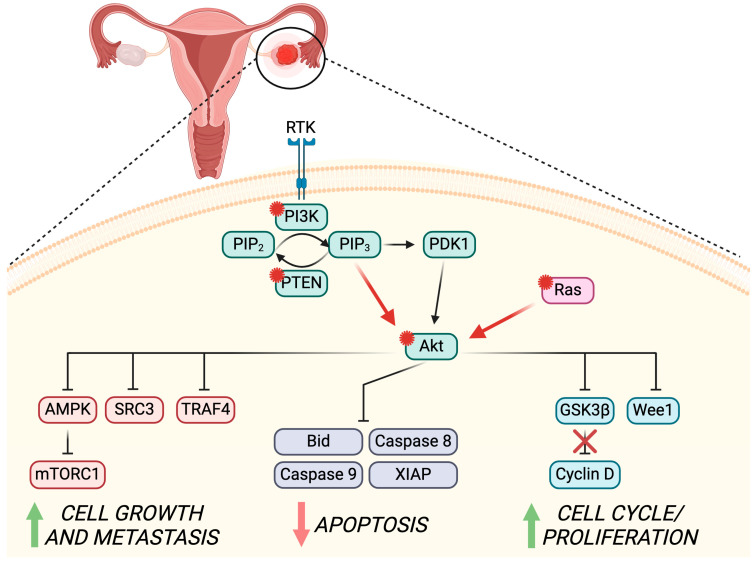
**PI3K/Akt dysregulation in ovarian cancer (OC)**. Akt, PI3K, and PTEN are the three main markers in the PI3K/Akt mechanisms that when mutated can initiate and promote OC. Pathogenesis and development of OC are predominantly supported by these markers via the dysregulation of key mechanisms, such as cell cycle and proliferation, apoptosis, and metastasis. The red asterisk (*) next to the protein name indicates oncogenic mutation. AMPK, AMP-activated protein kinase; Bid, BH3 interacting-domain death agonist; GSK3β, glycogen synthase kinase 3β; mTORC1, mammalian target of rapamycin complex 1; PI3K, phosphoinositide 3-kinase; PDK1, phosphoinositide 3-kinase-dependent kinase 1; PIP_2_, phosphatidylinositol-4,5-bisphosphate; PIP_3_, phosphatidylinositol-3,4,5-trisphosphate; PTEN, phosphatase and tensin homolog deleted on chromosome 10; Ras, rat sarcoma virus protein; RTK, receptor tyrosine kinase; SRC3, steroid receptor coactivator 3; TRAF4, tumor necrosis factor receptor-associated factor 4; XIAP, X-linked inhibitor of apoptosis protein. Created with BioRender.com.

## Data Availability

No new data were created or analyzed in this study. Data sharing is not applicable to this article.

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
