# Peer review of "An Update on Physiopathological Roles of Akt in the ReprodAKTive Mammalian Ovary"

_life, 2024, doi:10.3390/life14060722_

Round 1

Reviewer 1 Report

Comments and Suggestions for Authors

The role of AKT signaling pathway in ovary has been widely reviewed. The difference between this paper and other papers is that the research on ovarian cancer is added. Personally, I think it is not appropriate to include this part in this review.

Author Response

We would like to thank the reviewers for their comments, criticism, and suggestion. We took under consideration all of them and amended the manuscript at the best of our effort. We hope that the modifications made will be positively evaluated for publication in Life journal.

Sincerely,

Sandra Cecconi and all the authors

Reviewer 1

The role of AKT signaling pathway in ovary has been widely reviewed. The difference between this paper and other papers is that the research on ovarian cancer is added. Personally, I think it is not appropriate to include this part in this review.

Answer:

We agree with the reviewer regarding thepresence in literature of several reviews for the topic of Akt and the related signaling pathways in ovaries, female fertility and fertilization. To be transparent and aknowledge this, we citate the main reviews during the text and report them in the reference list (e.g., new references 1, 5, 3, 46). For this reason, we include the word update in the title. Nevertheless, as pointed out also by other reviewers, this manuscript is not only focused in reporting the results and literature on physiological functions of Akt signalling in the ovary, but also to describe how the imbalance of this pivotal pathway can give rise to ovarian pathological conditions related to fertility potential and health (i.e., PCOS and POF), and ovarian cancer. This essential part of our inclusive review is mistakenly missing from the title, therefore giving incomplete information about the content of the review. We therefore, modify the title as follows:

“An update on physiopathological roles of Akt in the reprodAKTive mammalian ovary”

We hope that this modification will be more inclusive and justify the presence of the pathological conditions, such as ovarian cancer, in the present review.

References:

  1. Cecconi, S.; Mauro, A.; Cellini, V.; Patacchiola, F. The Role of Akt Signalling in the Mammalian Ovary. International Journal of Developmental Biology 2012, 56, 809–817, doi:10.1387/ijdb.120146sc.
  2. Kalous, J.; Aleshkina, D.; Anger, M. A Role of PI3K/Akt Signaling in Oocyte Maturation and Early Embryo Development. Cells 2023, 12, 1830, doi:10.3390/cells12141830.
  3. De Felici, M.; Klinger, F.G. PI3K/PTEN/AKT Signaling Pathways in Germ Cell Development and Their Involvement in Germ Cell Tumors and Ovarian Dysfunctions. Int J Mol Sci 2021, 22, 9838, doi:10.3390/ijms22189838.
  4. Alcaráz, L.P.; Prellwitz, L.; Alves, G.; Souza-Fabjan, J.M.G.; Dias, A.J.B. Role of Phosphoinositide 3-Kinase/ Protein Kinase B/ Phosphatase and Tensin Homologue (PI3K/AKT/PTEN) Pathway Inhibitors during in Vitro Maturation of Mammalian Oocytes on in Vitro Embryo Production: A Systematic Review. Theriogenology 2022, 189, 42–52, doi:10.1016/J.THERIOGENOLOGY.2022.06.009.

Reviewer 2 Report

Comments and Suggestions for Authors

The review “ReprodAKTion: an update of Akt functions in the mammalian ovary” summarizes the importance of the PI3k/Akt pathway on ovarian development and addresses its role in the main ovarian pathologies. The manuscript is well written and describes an important area of development, with a broad approach. I leave some suggestions for minor revisions.

Title: The title seems very broad and suggests that topics related to pathologies will not be covered. The title could address the relationship between Akt and the pathologies mentioned throughout the review. I suggest changing the word ReprodATKtion.

Introduction: Throughout the review, aspects related to innovative assisted reproductive technologies are not covered in depth. I suggest that this information be removed from the introduction.

The authors could add a topic characterizing the PI3K/Akt pathway.

Conclusion: I believe that the authors can point out directions for future drug or therapy development studies using PI3K/AKT as a target based on topic 5.3.

Author Response

We would like to thank the reviewers for their comments, criticism, and suggestion. We took under consideration all of them and amended the manuscript at the best of our effort. We hope that the modifications made will be positively evaluated for publication in Life journal.

Sincerely,

Sandra Cecconi and all the authors

Reviewer 2

The review “ReprodAKTion: an update of Akt functions in the mammalian ovary” summarizes the importance of the PI3k/Akt pathway on ovarian development and addresses its role in the main ovarian pathologies. The manuscript is well written and describes an important area of development, with a broad approach. I leave some suggestions for minor revisions.

Title: The title seems very broad and suggests that topics related to pathologies will not be covered. The title could address the relationship between Akt and the pathologies mentioned throughout the review. I suggest changing the word ReprodATKtion.

Answer: We thank the reviewer for pointing out the mistakenly missing part from our title. Please, find below the modified title following your suggestions:

“An update on physiopathological roles of Akt in the repro-dAKTive mammalian ovary”.

Introduction: Throughout the review, aspects related to innovative assisted reproductive technologies are not covered in depth. I suggest that this information be removed from the introduction.

Answer: We agree with the reviewer and removed the part regarding assisted reproductive technologies from the introduction section.

The authors could add a topic characterizing the PI3K/Akt pathway.

Answer: We thank the reviewer for this suggestion, and we understand the valuable information connected to the characterization of PI3K/Akt pathway in health and disease. However, the present manuscript is entirely focused on this signaling in ovary physiology and pathology, thus being already fairly long. In order to avoid increasing the length of the manuscript, we amended the sentence in line 32-24 by adding a reference that reports the characterization of the pathway in both health and disease conditions, as follows:

“Among these, the phosphoinositide 3-kinase (PI3K)/ phosphatase and tensin homologue deleted on chromosome 10 (PTEN)/Akt and tuberous sclerosis (TSC)/mammalian target of rapamycin (mTOR) pathways have been extensively studied because they are critical regulators of ovarian folliculogenesis [1] and, in a broader perspective, of other pathophysiological mechanisms [2].”

References:

  1. Cecconi, S.; Mauro, A.; Cellini, V.; Patacchiola, F. The Role of Akt Signalling in the Mammalian Ovary. International Journal of Developmental Biology 2012, 56, 809–817, doi:10.1387/ijdb.120146sc.
  2. Hers, I.; Vincent, E.E.; Tavaré, J.M. Akt Signalling in Health and Disease. Cell Signal 2011, 23, 1515–1527, doi:10.1016/j.cellsig.2011.05.004.

Conclusion: I believe that the authors can point out directions for future drug or therapy development studies using PI3K/AKT as a target based on topic 5.3.

Answer: We thank the reviewer for this suggestion. We added in line 717-721 the following sentence:

“On these basis, development of novel drugs targeting different nodes of this pathway – e.g., eukaryotic translation initiation factor 4E (eIF4E), p70S6 kinase (p70-S6K), and pro-to-oncogene serine/threonine-protein kinase (PIM) – could help the progress of anti-OC therapies.”

Reviewer 3 Report

Comments and Suggestions for Authors

The review by Giaccari et al. focuses on the functions of Akt in the mammalian ovary. Overall, it is a well-written and interesting review focusing on one of the most relevant signaling pathways regulating the functions of the ovary. In addition, the authors highlighted the involvement of the pathway in pathological conditions. The following comments and suggestions are made to improve the review’s quality.

The review needs some work to fix some writing/grammatical issues. Paragraphs and sentences are too long. Some sentences lack references. Make sure you are using the proper gene nomenclature and protein names for human, rat and mouse. In addition, it is unclear if some conclusions or observations throughout the manuscript are based on clear evidence or if the authors are assuming based on indirect evidence.

Lines 21-24. Break in more sentences.

Lines 32-36. References

Lines 36-38. References

Line 50. Add “and” before “extensive epigenetic…”

Line 54. References

Line 68. References

Line 88. Define E7.5

Line 103. Define RTK

Line 146 inactivation? of who? Follicle? Should it be inactivation or activation here?

Line 256. The sentence starting in this line is not clear

Line 333. Odd space

Lines 333-335. Redundant use of “thus”. This reviewer’s suggestion is to delete the “thus” before “…indicating that the miR-18b-5p/PTEN”

Line 383. “Also”?

Line 394. PHLPP1 is not defined in the text

Line 446. Session or section?

Line 471. “This”?

The conclusion section has the same number (5) as the previous section

Homogenize abbreviations in subtitles. The authors defined PCOS in subtitle 4.1 but you did not define POF in subtitle 4.2

Comments on the Quality of English Language

Minor editing of English language required

Author Response

We would like to thank the reviewers for their comments, criticism, and suggestion. We took under consideration all of them and amended the manuscript at the best of our effort. We hope that the modifications made will be positively evaluated for publication in Life journal.

Sincerely,

Sandra Cecconi and all the authors

Reviewer 3

The review by Giaccari et al. focuses on the functions of Akt in the mammalian ovary. Overall, it is a well-written and interesting review focusing on one of the most relevant signaling pathways regulating the functions of the ovary. In addition, the authors highlighted the involvement of the pathway in pathological conditions. The following comments and suggestions are made to improve the review’s quality.

The review needs some work to fix some writing/grammatical issues. Paragraphs and sentences are too long. Some sentences lack references. Make sure you are using the proper gene nomenclature and protein names for human, rat and mouse. In addition, it is unclear if some conclusions or observations throughout the manuscript are based on clear evidence or if the authors are assuming based on indirect evidence.

Lines 21-24. Break in more sentences.

Answer: We thank the reviewer for this suggestion. We amended and divide the sentence in lines 21-25 as follows:

“Due to the pivotal role in the control of cell proliferation, apoptosis, and survival mechanisms, the dysregulation of this molecular pathway can trigger onset of pathological conditions. Among those, we will focus on disease that can harm female fertility, as polycystic ovary syndrome and premature ovarian failure, or women’s general health, as ovarian cancer.”

Lines 32-36. References

Answer: We added the references 1 and 2, as requested by the reviewer.

References:

  1. Cecconi, S.; Mauro, A.; Cellini, V.; Patacchiola, F. The Role of Akt Signalling in the Mammalian Ovary. International Journal of Developmental Biology 2012, 56, 809–817, doi:10.1387/ijdb.120146sc.
  2. Hers, I.; Vincent, E.E.; Tavaré, J.M. Akt Signalling in Health and Disease. Cell Signal 2011, 23, 1515–1527, doi:10.1016/j.cellsig.2011.05.004.

Lines 36-38. References

Answer: We added the references 1 and 3, as requested by the reviewer.

References:

  1. Cecconi, S.; Mauro, A.; Cellini, V.; Patacchiola, F. The Role of Akt Signalling in the Mammalian Ovary. International Journal of Developmental Biology 2012, 56, 809–817, doi:10.1387/ijdb.120146sc.
  2. Kalous, J.; Aleshkina, D.; Anger, M. A Role of PI3K/Akt Signaling in Oocyte Maturation and Early Embryo Development. Cells 2023, 12, 1830, doi:10.3390/cells12141830.

Line 50. Add “and” before “extensive epigenetic…”

Answer: Amended.

Line 54. References

Answer: We added the reference 5, as requested by the reviewer.

Reference:

  1. De Felici, M. Origin, Migration, and Proliferation of Human Primordial Germ Cells. In Oogenesis; Coticchio, G., Albertini, D.F., De Santis, L., Eds.; Springer London: London, 2013; pp. 19–37 ISBN 978-0-85729-826-3.

Line 68. References

Answer: We added the reference 11, as requested by the reviewer.

Reference:

  1. Depmann, M.; Faddy, M.J.; van der Schouw, Y.T.; Peeters, P.H.M.; Broer, S.L.; Kelsey, T.W.; Nelson, S.M.; Broekmans, F.J.M. The Relationship Between Variation in Size of the Primordial Follicle Pool and Age at Natural Menopause. J Clin Endocrinol Metab 2015, 100, E845–E851, doi:10.1210/jc.2015-1298.

Line 88. Define E7.5

Answer: Amended.

Line 103. Define RTK

Answer: Amended.

Line 146 inactivation? of who? Follicle? Should it be inactivation or activation here?

Answer: Amended.

Line 256. The sentence starting in this line is not clear

Answer: Amended.

Line 333. Odd space

Answer: Amended.

Lines 333-335. Redundant use of “thus”. This reviewer’s suggestion is to delete the “thus” before “…indicating that the miR-18b-5p/PTEN”

Answer: Amended.

Line 383. “Also”?

Answer: Amended.

Line 394. PHLPP1 is not defined in the text

Answer: Amended.

Line 446. Session or section?

Answer: Amended.

Line 471. “This”?

Answer: Amended.

The conclusion section has the same number (5) as the previous section

Answer: Amended.

Homogenize abbreviations in subtitles. The authors defined PCOS in subtitle 4.1 but you did not define POF in subtitle 4.2

Answer: We removed the abbreviation from the subtitle 4.1 and checked the abbreviations through the whole manuscript.

Reviewer 4 Report

Comments and Suggestions for Authors

The authors had made great efforts to examine these pertinent molecular pathways to ovarian lifespan and functions which is highly commendable.
Minor points to consider:

1. The tables for the purported actions and links to how the molecular pathways work are very important but there should be another column to determine if the work is derived from animal studies or human studies. Many animal models of PCOS and POF are induced and may not mimic the true phenomenon observed in patients.

2. Can we only include human studies on the purported pathways and then assessed if the findings are then tested in animal models? This makes the manuscript more focused and direct human relevance. The induced animal models may skew the metabolic functions such as proposing hyperandrogenism as the inevitable cause for PCOS only.

Author Response

We would like to thank the reviewers for their comments, criticism, and suggestion. We took under consideration all of them and amended the manuscript at the best of our effort. We hope that the modifications made will be positively evaluated for publication in Life journal.

Sincerely,

Sandra Cecconi and all the authors

Reviewer 4

The authors had made great efforts to examine these pertinent molecular pathways to ovarian lifespan and functions which is highly commendable.

Minor points to consider:

  1. The tables for the purported actions and links to how the molecular pathways work are very important but there should be another column to determine if the work is derived from animal studies or human studies. Many animal models of PCOS and POF are induced and may not mimic the true phenomenon observed in patients.

Answer: We thank the reviewer for this suggestion. We amended the tables 1 and 2 as suggested.

  1. Can we only include human studies on the purported pathways and then assessed if the findings are then tested in animal models? This makes the manuscript more focused and direct human relevance. The induced animal models may skew the metabolic functions such as proposing hyperandrogenism as the inevitable cause for PCOS only.

Answer: We agree with the reviewer with the importance of human relevant studies to avoid the bias encountered during animal studies. As the reviewer points out, the induced disease (either PCOS or POF) on animals is not always true to the actual mechanism and physiopathology of humans. Nevertheless, the direct study of ovarian microenvironment in these pathological conditions is seldomly possible in humans because it require the invasive retrieval of ovarian biopsies. As a consequence, the direct studies on human are often based on endocrinological and cell targeted approaches (i.e., granulosa cells isolated after follicular fluid aspiration during ART procedures). Therefore, we believe that including the animal studies can provide great insights on ovarian-specific consequences of PCOS and POF, taking under consideration all the limitations of these data and the difficulty in translating the results on women. We reorganized the text in sections 4.1. Polycistic ovary syndrome, 4.2. Premature ovarian failure and 4.3. Potential treatments for reproductive pathologies involving the PI3K/Akt pathway. In detail, we report and discuss the relevant data from human-based studies, and subsequently introduced the animal studies as complementary information.

Round 2

Reviewer 1 Report

Comments and Suggestions for Authors

None